# Psychometric Properties of the Italian Version of Sensory Processing and Self-Regulation Checklist (SPSRC)

**DOI:** 10.3390/healthcare11010092

**Published:** 2022-12-28

**Authors:** Giulia Purpura, Cynthia Y. Y. Lai, Giulia Previtali, Ivan Neil B. Gomez, Trevor W. K. Yung, Luca Tagliabue, Francesco Cerroni, Marco Carotenuto, Renata Nacinovich

**Affiliations:** 1School of Medicine and Surgery, University of Milano Bicocca, 20900 Monza, Italy; 2Department of Rehabilitation Sciences, The Hong Kong Polytechnic University, Kowloon, Hong Kong SAR, China; 3Department of Occupational Therapy, University of Santo Tomas, Manila 1015, Philippines; 4Child and Adolescent Health Department, San Gerardo Hospital, ASST of Monza, 20900 Monza, Italy; 5Clinic of Child and Adolescent Neuropsychiatry, Università degli Studi della Campania “Luigi Vanvitelli”, 81100 Caserta, Italy

**Keywords:** sensory processing, sensory integration, self-regulation, neurodevelopmental disabilities, children

## Abstract

Sensory processing abilities play important roles in child learning, behavioural and emotional regulation, and motor development. Moreover, it was widely demonstrated that numerous children with neurodevelopmental disabilities show differences in sensory processing abilities and self-regulation compared with those of typical children. For these reasons, a complete evaluation of early symptoms is very important, and specific tools are necessary to better understand and recognize these difficulties during childhood. The main aim of this study was to translate, culturally adapt, and validate in a population of Italian typically developing (TD) children the Sensory Processing and Self-Regulation Checklist (SPSRC), a 130-item caregiver-reported checklist, covering children’s sensory processing and self-regulation performance in daily life. Preliminary testing of the SPSRC-IT was carried out in a sample of 312 TD children and 30 children with various developmental disabilities. The findings showed that the SPSRC-IT had high internal consistency, a good discriminant, and structural and criterion validity about the sensory processing and self-regulation abilities of children with and without disabilities. These data provide initial evidence on the reliability and validity of SPSRC-IT, and the information obtained by using the SPSRC-IT may be considered a starting point to widen the current understanding of sensory processing difficulties among children.

## 1. Introduction

Learning processes are the most visible results of sensory integration and self-regulation capacities in childhood. Sensory processing and integration refer to the abilities of detecting, modulating, perceiving, discriminating, and using several sensory information from different sensory channels; self-regulation is the ability of the individual to independently organize environmental experiences, to regulate his/her internal emotional state, and to respond with appropriate behaviour to the external context. Thanks to these early neurodevelopmental functions, it is possible to gain gradual awareness of the body and better knowledge of environmental elements to adapt throughout daily life. Moreover, sensory processing abilities develop and play important roles in child learning, behavioural and emotional regulation, motor development, and task performance [1]. In this regard, sensory processing and self-regulation are strictly interlinked, and their maturation are reciprocally connected, especially during the first years of life.

De Gangi and colleagues [2] deeply studied these two aspects of neurodevelopment and suggested that children initially detected with these types of early dysfunctions are at high risk for later perceptual, language, and emotional/behavioural difficulties in the preschool and school years. The Diagnostic Classification of Mental Health and Developmental Disorders of Infancy and Early Childhood (DC: 0.5) [3] provides criteria for the diagnosis of sensory processing disorders (SPD) that can be used when the infant/young child demonstrates atypical behaviours that are believed to reflect abnormalities in regulating sensory inputs, with a profound impact on his/her functioning during daily activities. Recently, Chien et al. [4] reported that children with potential SPD have important constraints in the degree of participation and enjoyment during daily life in comparison with those of typical children. For these reasons, a complete evaluation of early symptoms (and how these affect functional performances) is very important, and specific tools are necessary to better understand and recognize these types of difficulties during childhood. Moreover, it is well-known that the possible developmental trajectories of children with early SPD can be many and very different, and that also a lot of children with neurodevelopmental disabilities show differences in sensory processing abilities in comparison with those of typical children, for example, children affected by autism spectrum disorders (ASD), developmental coordination disorders (DCD), or attention deficit hyperactivity disorders (ADHD) [5,6,7]. Children who exhibit difficulties in sensory processing may have difficulties in regulating the sleep–wake rhythm, in maintaining a state of positivity during interactions with other people, or in sustaining attention at appropriate levels for learning; or they may over-react to daily environmental stimuli, with problems in modifying their behaviour for normal daily routines, with a negative impact on their adaptation to the environment.

In this context, Lai and collaborators [8,9] recently developed and implemented a new parent/caregiver-answered questionnaire that incorporates both aspects of sensory processing and aspects of self-regulation, called the Sensory Processing and Self-Regulation Checklist (SPSRC). It can be used in children with and without disabilities, aged 3–8 years. The SPSRC is a single instrument that comprises two parts: self-regulation abilities and sensory processing skills. The former focus on children’s difficulties with behavioural regulation and comprises items about physiological conditions and social, emotional, and cognitive development, while the latter is aimed at identifying behavioural responses to sensory stimuli and at quantifying the degree of difficulty of children upon receiving different types of sensory stimulations. The original SPSRC items are written in HK-Chinese, but the checklist was translated and validated in English through a multiple-step process of language equivalency. The good reliability and validity and the good discriminatory capacity of SPSRC in examining sensory processing and self-regulation in typical and atypical children was demonstrated in both versions [10,11]. More recently, Gomez and collaborators have also performed the linguistic and cultural equivalency of the SPSRC-Tagalog for the assessment of sensory processing and self-regulation abilities of Tagalog-speaking Filipino children, and its psychometrics properties also in this case are retained [12]. The main aim of this study was to translate, culturally adapt, and validate the SPSRC in a population of typically developing (TD) Italian children by measuring internal consistency and cross-cultural validity. The questionnaire was also administered to a group of caregivers of children with developmental disabilities (DD) with the aim of controlling the discriminatory capacity of the SPSRC.

## 2. Materials and Methods

The developers of the original SPSRC and of the SPSRC-English Version agreed to translate and culturally adapt the tool into Italian. Thus, SPSRC was translated from English to Italian by the researchers and clinicians of the University of Milano-Bicocca using the Translation and Cultural Adaptation of Patient Reported Outcome Measures–Principles of Good Practice guidelines [13]. We performed an observational and cross-sectional study, and data were collected through an online form (see Section 2.2).

### 2.1. Translation and Cultural Adaptation

Two Italian undergraduate students in neurodevelopmental disorders therapy, familiar with the English language, independently forward-translated the SPSRC-English Version [14] into Italian. The two preliminary forward translations were synthesized in one single Italian version by the monitoring and revision of one Italian neurodevelopmental therapist experienced with sensory integration and familiar with the English language. A native English speaker (expert teacher in primary and middle school classes of an English school in Italy) performed a back-translated version (from Italian to English), and then this English version was compared with the original version and approved by the working group. To adapt the translated version to Italian culture, a team of neurodevelopmental disorders therapists who were familiar with both languages reviewed the preliminary translated version and then developed the final version. To check understandability, the final version was shown to three parents (with children between 3 and 8 years old) who were not familiar with sensory integration terminology. Some items in the English and Italian Versions of SPSRC are reported in Table 1. In the Italian version, five items (four items in Part 1 and one item in Part 2) were removed (see Section 3.2).

### 2.2. Participants and Procedures

Parents of children between 3 and 8.11 years old were recruited for the study through Italian kindergartens and primary schools and through social networks and online forums. The questionnaire was completed using an online self-administered form (developed with Google modules [Mountain View, CA, USA]). Parents intent in participating were informed about the modalities and purposes of the study. The entire compilation was completely anonymous. The first part of form reported the aim of the study and provided a general questionnaire (GQ) that permitted the collection of the consensus and of the main sociodemographic data of families: residence (region of Italy), native language of family, age of the child, type of relationship of the caregiver with the child, attended school class by the child, gestational age and weight at birth of the child, and whether or not there was a neuropsychiatric diagnosis in the child (if yes, the type of diagnosis). In the second part of online form, two questionnaires were proposed (see Measures Section). Recruitment was performed from 18 May 2022 to 15 November 2022.

### 2.3. Measures

#### 2.3.1. Sensory Processing and Self-Regulation Checklist (SPSRC)

The SPSRC was developed and validated in Chinese [8,9] and then translated and validated in English [14] as a single checklist that provides information on both sensory processing and self-regulation for children from 3 to 8 years of age. Good psychometric properties have been reported for the SPSRC in both versions [10,11]. The checklist is composed of two parts, subdivided into several sections and factor scales. Part 1 (37 items) investigates self-regulation and has three sub-sections [(A) Physiological Conditions, (B) Social/Cognitive/Emotional Development, and (C) Behaviours When Facing Changes or Challenges] and four factor scales [(1) Emotional Regulation, Facing Challenges; (2) Emotional Regulation, Facing Changes; (3) Physiological Regularity and Response to Soothing; and (4) Autonomic Activity]. Part 2 (93 items) consists of items related to sensory processing and is further divided into six sub-sections [(A) Auditory Sense, (B) Visual Sense, (C) Tactile Sense, (D) Gustatory and Olfactory Sense, (E) Vestibular Sense, and (F) Proprioceptive Sense] and four factors [(1) Sensory-Seeking Behaviour, (2) Sensory Under-Responsivity, (3) Sensory Over-Responsivity, and (4) Stability of Sensory Responsivity]. The parents were instructed to report on their child’s typical performance within the last three months for the items on the checklist using a 5-point Likert scale (5 = never, 4 = seldom, 3 = sometimes, 2 = frequently, and 1 = always); some items had opposite polarity (thus, a reversed scoring). A higher score indicated a more favourable performance (fewer negative symptoms).

#### 2.3.2. Short Sensory Profile-2nd Version (SSP-2)

The Short Sensory Profile 2 [15] is a 34-items parent questionnaire designed to measure behaviours associated with abnormal responses to sensory stimuli in children aged 3–14.11 years. The SSP-2 is a reliable and valid tool used to look at children’s sensory processing issues that affect their performance, and it was already translated, adapted, and validated for the Italian population [16]. The SSP-2 provides scores in the four quadrants of Dunn’s Sensory Processing Framework, based on the child’s neurological threshold to sensory input and their reactive behaviours (Seeking, Sensitivity, Avoiding, and Registration) [17]. The parents/caregivers evaluated the child on each item using a 5-point Likert scale (5 = almost always, 4 = frequently, 3 = half the time, 2 = occasionally, 1 = almost never).

### 2.4. Statistical Analysis

Anonymous data from the participants were initially encoded in MS Excel. Successively, the IBM^®^ SPSS^®^ 28.0 tool (Chicago, IL, USA) was used to carry out the statistical analysis. The psychometric properties of the SPSRC-IT were evaluated according to the Consensus-Based Standards for the Selection of Health Status Measurement Instrument (COSMIN) checklist [18]. A *p*-value less than or equal to 0.05 was considered statistically significant.

The first step was testing the *structural*, *discriminant,* and *known-group validities*. *Structural validity* included the defining of the structure and factors of checklist on the basis of the original version [10], the calculations of the Kaiser–Meyer–Olkin value and of Bartlett’s sphericity test, the analysis of the content of items and item reduction (if necessary), and the evaluation of the construct by a confirmatory factor analysis. *Discriminant validity* was examined using Pearson’s correlation coefficient between the mean scores of the two parts of the questionnaire. *Known-group validity* was analyzed by comparing, with an independent T-test, the group differences based on age (pre-schoolers from 3 to 6.6 years and school-aged from 6.7 to 8.11 years), gender, and disability status, considering the parts, subscales, factors, and composite scores of the SPSRC-IT.

The second step was the verification of *reliability* by performing *internal consistency*. The internal consistency of the SPSRC-IT was examined by Cronbach’s alpha. Alpha values of 0.7, 0.8, and 0.9 are thought to represent a fair, good, and excellent degree of internal consistency, respectively.

Finally, the *criterion validity* was examined with *concurrent validity* testing by comparing the relevant parts, subscales, and total scores of SPSRC-IT with the Italian Version of SSP-2 for the whole sample and only for the DD children (clinical group).

## 3. Results

### 3.1. Participant Demographics

The SPSRC-IT was performed by 388 caregivers. Of these, 46 were excluded for lack of data or due to the age of child being out of the range of the study. Thus, data for the analysis involved 342 preschool and school-aged children (312 with typical development (TD) and 30 with developmental disabilities (DD)), with a mean age of 5.7 years (SD: 1.7). In the TD group, there were slightly more boys (52.6%) than girls (47.4%), while in the DD group, 73.3% were boys and 26.7% were girls. In Table 2, the demographics data of the participants are reported. The specific diagnoses of the DD children varied, and they are reported in Table 3.

### 3.2. Structural Validity

To evaluate the construct validity of the SPSRC-IT, the confirmatory factor analysis (CFA) was used, considering the whole sample (TD and DD children). With the CFA, we tested whether or not the data fit the hypothesized factors model in the original HK-Chinese and English SPSRC.

First, preliminary tests were performed, confirming the adequacy of sample: the Kaiser–Meyer–Olkin statistic was 0.83, which indicated that the data were meritorious for the factors analysis, while Bartlett’s test of sphericity was statistically significant (*p* < 0.001), showing that the correlation matrix was not an identity matrix (rejection of the null hypothesis). Thus, the variables were adequate for factor analysis.

CFA was carried out after the analysis of the contents of items by a panel of neurodevelopmental therapists and on the basis of the number of items for each factor, according to the Chinese original version by Lai et al. [10].

Part 1 of the SPSRC-IT considered four factors in the CFA. On the basis of this, the final version of Part 1 of the SPSRC-IT includes 33 items (vs. the 37 items of original version): Factor 1, “Emotional Regulation–Facing Challenges”, includes 10 items (factor loading range 0.44–0.80) and reflects a child’s emotional regulation abilities when facing challenging situations; Factor 2, “Emotional Regulation–Facing Changes”, includes 11 items (factor loading range 0.36–0.76) and reflects a child’s emotional regulation abilities when facing changes in routines or events; Factor 3, “Physiological Regularity and Response to Soothing”, includes 6 items (factor loading range 0.36–0.85) and reflects a child’s physiological patterns (e.g., sleeping, bladder, and bowel patterns) and responses to calming by adults; Factor 4, “Autonomic Activity”, includes 6 items (factor loading range 0.36–0.60) and reflects the child’s autonomic responses (e.g., palm sweating). Because four items of the Italian version had low estimations (<0.35), they were removed.

Additionally, for Part 2 of the SPSRC-IT, on the basis of the contents of items and on the factors of the original version of the checklist by Lai et al. [10], a CFA on a four-factor solution was performed. Factor 1, “Sensory-Seeking Behaviour”, includes 34 items (factor loading range 0.42–0.81) and reflects a child’s tendency to demonstrate behaviours related to seeking sensory stimuli in the surrounding environment. Factor 2, “Sensory Under-Responsivity”, includes 23 items (factor loading range 0.59–0.90) and reflects a child’s diminished or lack of response to sensory stimulation. Factor 3, “Sensory Over-Responsivity”, includes 29 items (factor loading range 0.60–0.85) and reflects a child’s exaggerated response toward sensory stimulation. Factor 4, “Stability of Sensory Responsivity”, comprises six items (factor loading range 0.82–0.89) and reflects a child’s stability in response to sensory stimulation across situations. Only one item was removed for a low estimation (<0.35); thus, the second part of the finalized SPSRC-IT comprises 92 items (vs. the 93 items of the original version).

### 3.3. Discriminant Validity

The discriminant validity was examined by testing the relationship between the mean scores of Part 1 (self-regulation abilities, 33 items) and Part 2 (sensory processing abilities, 92 items) of the SPSRC-IT. The two-tail Pearson correlation coefficient was r = 0.621 (*p* < 0.001).

### 3.4. Known-Group Validity

Data of the 312 TD children from different age groups (213 pre-schoolers vs. 99 school-aged) were compared on their SPSRC-IT mean scores (parts, subscales, factors, and total) using an independent T-Test. The results showed significantly different SPSRC-IT scores across age groups, specifically for Part 1 Self-Regulation Abilities (*p* = 0.011, t = −2.579), for Subscale 1A Physiological Conditions (*p* = 0.029, t = −2.205), for Subscale 1C Behaviours When Facing Changes or Challenges (*p* = 0.002, t = −3.136), for Factor 1 of Part 1 Emotional Regulation–Facing Challenges (*p* = 0.006, t = −2.781), for Subscale 2A Auditory Sense (*p* = 0.012, t = −2.549), and for Subscale 2C Tactile Sense (*p* = 0.026, t = −2.247).

Gender differences (164 boys vs. 148 girls) in the SPSRC mean scores of TD children (parts, subscales, factors, and composite) were likewise compared, and T-test results revealed statistical differences only in the subscale 1B Social/Cognitive/Emotional Development (*p* = 0.020, t = −2.348), 2A Auditory Sense (*p* = 0.010, t = −2.581), 2B Vision Sense (*p* = 0.045, t = −2.012), Factor 2 of Part 1 Emotional Regulation–Facing Changes (*p* = 0.006, t = −2.759), and Factor 4 of Part 2 Stability of Sensory Responsivity (*p* = 0.036, t = −2.112), in which girls had better scores than boys.

Finally, we performed a comparison between TD children (*n* = 312) and a sample of DD children (*n* = 30) in the mean scores of SPSRC-IT (parts, subscales, factors, and total), using an independent T-test (see Table 4). The results indicated significant differences in Part 1 Self-Regulation Abilities (*p* < 0.001, t = 5.840), Part 2 Sensory Processing Abilities (*p* < 0.001, t = 4.784), in Total Score (*p* < 0.001, t = 5.603), and in all subscales and factors. Globally, TD children showed higher scores than DD children, indicative of more favourable performances.

### 3.5. Internal Consistency

The consistency of responses to the items of the SPSRC-IT was tested to determine whether or not each of the subscale and composite scores measured the same general construct. Cronbach’s α coefficients were 0.89 for the items of Part 1 (33 checklist items) and 0.95 for the items of Part 2 (92 checklist items) of the SPSRC -IT. Overall, Cronbach’s α coefficient for the SPSRC composite (125 checklist items) was 0.95.

### 3.6. Concurrent Validity

The mean scores (parts, subscales, factors, and composite) of the whole sample and of the DD children on their SPSRC-IT were analyzed for their concurrent relationship to their respective scores on the SSP-2. Both in the whole sample and in the DD group, the Pearson correlation analysis showed strong negative correlations between the several scores of SPSRC-IT and the various scores of SSP-2 (Total Sample: SPSRC Total Score vs. SSP-2 Total Score, *p* < 0.001, rho = −0.652; DD Group: SPSRC Total Score vs. SSP-2 Total Score, *p* < 0.001, rho = −0.697). The presence of negative correlations was justified by the inverse modality of scores in the two tools.

## 4. Discussion

The main purpose of this study was to examine the psychometric properties of the SPSRC-IT. The several evaluation methods used determined that the SPSRC-IT had overall good psychometric properties in measuring the sensory processing and self-regulation skills of children with and without disabilities, with an age range from 3 to 8.11 years. This research continues the path of the earlier findings from the original version of the SPSRC (HK-Chinese) for children from 3 to 8 years old [11] and from the English version of the SPSRC for children from 4 to 12 years old [10]. Taken as a whole, the parts (Parts 1 and 2) and composite (overall ability) scores of the SPSRC-IT have excellent internal consistency, similar both with the original version in Chinese and with the English version. This means that the items of the SPSRC-IT reliably measure the capacities of regulation and behavioural responses to sensory stimuli that a child may encounter in daily life activities, in the Italian context. Similarly, the construct validity of the SPSRC-IT appears very good, indicating that in each part, subscale, and factor, the checklist is able to discriminate the underlying constructs, to measure several dimensions of self-regulation and sensory processing, and to detect similarities and differences between different groups of children. However, it is important to report that some slight differences are present between the previous versions and the Italian version of the SPSRC. As a matter of fact, in the current Italian SPSRC, 125 items seem sufficient to score the self-regulation and sensory processing abilities in children, instead of the 130 items of the original HK-Chinese version.

Interesting data also emerged from analysis for the known-group validity. It is widely known that self-regulation plays a critical role in the development of children’s well-being, and several factors may affect a child’s behaviour during daily activities. Based on this, we tested three relevant variables (age, gender, and disability status) to understand if the SPSRC-IT can also identify subtle differences in different populations. First, there are some significant differences, especially in self-regulation, among children of preschool age (from 3 to 6.6 years) and school-aged (from 6.7 to 8.11 years). This finding is in line with previous studies that suggested long and complex maturation trajectories, both with regard to self-regulation abilities [19] and with regard to sensory integration processes [20,21,22], between 3 and 8 years. As a matter of fact, the maturation of attention regulatory functions in response to stimuli from the surrounding environment follows the development of emotion regulation and also vice versa; thus, during the preschool years, children are gradually more able to internalize the behavioural expectations imposed by their environments because they can perform more complicated tasks that require more ability to process sensory stimuli [23]. Nevertheless, these processes are very long and evolve thanks to the relationships with other people, in school, across life transitions, and throughout their lifespan [24]. Moreover, slight gender differences were found in the self-regulation and sensory processing abilities of the participants in this study. This finding confirms earlier results, which showed that boys and girls could have some differences in trajectories of the maturation of self-regulation abilities, although this point is greatly influenced by the environment and by models imposed by parents, cultures, and society [25,26]. For these reasons, further and more specific studies are necessary to better understand this aspect.

Similar to the English version of the SPSRC, TD were compared with children with disabilities, and important differences in all scores were found between the two groups, particularly since the DD group had significantly lower scores compared with those of typically developing children on their self-regulation, sensory processing, and overall abilities. Additionally, this finding is in line with the scientific literature, which reported frequent sensory processing issues in children with different types of disabilities [27,28,29,30]. This point is very important, because, contrary to other tools, the SPSRC takes into consideration the capacity of self-regulation, including physiological functions, that may impact negatively on a sensory processing disorder or may be consequently impaired in a sensory processing disorder. Thus, for both children with or at risk for developmental disabilities, the SPSRC-IT could be a useful instrument for planning tailored interventions and measuring the effects of treatment over the time.

Finally, the numerous scores on the SPSRC-IT show high correlations to the different domains and scales of the SSP-2, similar to the original SPSRC and to the English version of the SPSRC, in which the authors reported good convergent validities with respect to the Chinese and English SSP. This finding confirmed how well scores on the SPSRC-IT adequately reflect those of a proven measure. With the validation of this tool, it will be possible not only to define the mutual influence between sensory problems and self-regulation behaviours but also to promote cooperation between neurodevelopmental therapists, paediatricians, and psychologists, according to the principles of family-centred care, for a better planning and monitoring of early rehabilitation interventions. In addition to these clinical advantages, it will now be possible to compare the results obtained from studies conducted in Italy with those conducted in Chinese, Tagalog, or English-speaking countries, contributing to research in this area.

## 5. Conclusions

In conclusion, this study provides evidence supporting the positive psychometric properties of the Italian version of the SPSRC. In collaboration with the authors of the original versions of this checklist, we wanted to extend the previous line of research on the SPSRC and provide further evidence about its internal consistency and its usefulness both in clinical and research application areas. SPSRC-IT proved to have excellent reliability, a good structural construct, and adequate criterion validity for the assessment of sensory processing and self-regulation abilities in children aged 3–8.11 years old, with and without disabilities. Moreover, the information obtained from the SPSRC-IT may be a starting point to increase the current understanding of sensory processing difficulties among children.

## Figures and Tables

**Table 1 healthcare-11-00092-t001:** Example of English items translated into Italian language.

Part 1: Self-Regulation	Examples of English Items	Examples of Italian Items
Section 1A: *Physiological Condition*	Has good sleep quality, can sleep till morning (or falls asleep again 10 min after waking)	Ha una buona qualità del sonno, riesce a dormire fino al mattino (o si riaddormenta entro dieci minuti dal risveglio)
Section 1B: *Social/Cognitive/Emotional* *Development*	Unable to comprehend adults’ intentions or requests by observing their facial expressions, gestures, body languages or speeches	È incapace di comprendere le intenzioni o le richieste degli adulti osservando il loro viso, espressioni, gesti, linguaggi del corpo o discorsi
Section 1C: *Behaviours When Facing Changes or Challenges*	Becomes nervous and tenses up, movements become stiff (e.g., head and trunk turn together, hands and feet appear to be nailed on the ground) in new environment or when facing challenges	In un nuovo ambiente o di fronte a sfide, diventa nervoso e si irrigidisce, i movimenti diventano rigidi (ad es. testa e tronco ruotano insieme, mani e piedi sembrano inchiodati a terra)
**Part 2: Sensory Processing**		
Scale 2A: *Auditory Sense*	Unable to notice or shows no response to low volume sounds (e.g., sounds produced by musical boxes, whispers, or gentle door knocking sounds)	È incapace di notare o non mostra alcuna risposta a suoni di basso volume (ad es. prodotti da carillon, sussurri o leggero bussare sulla porta)
Scale 2B: *Visual Sense*	Seeks visual stimulations by staring at changing lights for a long time (e.g., TV screens or Christmas lights)	Cerca stimoli visivi fissando a lungo luci che cambiano (ad es. schermi Tv o luci natalizie)
Scale 2C: *Tactile Sense*	Appears excessively nervous, distressed, or makes complaints when hands or face are made wet by rain or splashes of water	Appare eccessivamente nervoso, angosciato o si lamenta quando le mani o il viso sono bagnate dalla pioggia o dagli schizzi d’acqua
Scale 2D: *Gustatory & Olfactory Sense*	Sniffs before manipulating objects or playing with toys	Annusa prima di manipolare oggetti o giocare con i giochi
Scale 2E: *Vestibular Sense*	Unable to notice or shows no response when he/she is about to fall	È incapace di notare o non mostra alcuna risposta quando sta per cadere
Scale 2F: *Proprioceptive Sense*	Likes to walk on tiptoes	Gli piace camminare in punta di piedi

**Table 2 healthcare-11-00092-t002:** Socio-demographic information of the participants.

		Typical Children (*n* = 312)	Children with Disabilities (*n* = 30)
**Gender**	*Boys*	164 (52.6%)	22 (73.3%)
	*Girls*	148 (47.4%)	8 (26.7%)
**Residence**	*North*	194 (62.2%)	18 (60%)
	*Center*	53 (17%)	5 (16.7%)
	*South*	65 (20.8%)	7 (23.3%)
**Caregiver**	*Mothers*	270 (86.5%)	26 (86.7%)
	*Fathers*	13 (4.2%)	4 (13.3%)
	*Both*	13 (4.2%)	0
	*Others*	16 (5.1%)	0
**Age**	*Mean*	5.7 years	6.2 years
	*Range*	3–8.11 years	3.3–8.11 years
	3 years	63 (20.2%)	4 (13.3%)
	4 years	63 (20.2%)	5 (16.7%)
	5 years	55 (17.6%)	5 (16.7%)
	6 years	54 (17.3%)	4 (13.3%)
	7 years	39 (12.5%)	4 (13.3%)
	8 years	38 (12.2%)	8 (26.7%)
*Total n (%)*		*312 (100%)*	*30 (100%)*

**Table 3 healthcare-11-00092-t003:** Summary of disabilities of the DD sample.

Types of Disabilities	
Autism spectrum disorder	10 (33.3%)
Language/communication disorders	5 (16.7%)
Regulation/behavioural disorders	4 (13.3%)
Neurologic disorders (i.e., cerebral palsy, epilepsy, etc.)	4 (13.3%)
Learning disorders	2 (6.7%)
Developmental coordination disorder	2 (6.7%)
Genetic disorders	2 (6.7%)
Intellectual disability	1 (3.3%)
*Total*	*30 (100%)*

**Table 4 healthcare-11-00092-t004:** Comparison of the scores of SPSRC-IT between TD and DD children.

Scores	TD Group Mean (SD)	DD Group Mean (SD)	*p*-Value
**Total Score**	555.6 (34.4)	502.7 (50.7)	**<0.001**
**Part 1: Self-Regulation**	140.1 (11.5)	121.7 (16.9)	**<0.001**
Section 1A: Physiological Conditions	35.04 (3.1)	32.3 (4.8)	<0.001
Section 1B: Social/Cognitive/Emotional Development	52.2 (5.9)	43.8 (8.1)	<0.001
Section 1C: Behaviours When Facing Changes or Challenges	52.9 (5.1)	45.5 (6.8)	<0.001
Factor 1: Emotional Regulation–Facing Challenges	45.3 (4.2)	39.4 (6.0)	<0.001
Factor 2: Emotional Regulation–Facing Changes	44.3 (5.1)	36.7 (6.2)	<0.001
Factor 3: Physiological Regularity and Response to Soothing	24.1 (3.3)	20.9 (4.9)	<0.001
Factor 4: Autonomic Activity	26.4 (2.4)	24.7 (3.3)	0.013
**Part 2: Sensory Processing**	415.5 (26.7)	381.03 (38.7)	**<0.001**
Scale 2A: Auditory Sense	69.3 (5.2)	63.1 (8.9)	<0.001
Scale 2B: Vision Sense	61.9 (3.5)	57.5 (6.1)	<0.001
Scale 2C: Tactile Sense	89.7 (5.9)	83.6 (8.2)	<0.001
Scale 2D: Gustatory and Olfactory Sense	56.6 (4.1)	51.9 (6.9)	<0.001
Scale 2E: Vestibular Sense	77.3 (7.1)	70.9 (9.0)	<0.001
Scale 2F: Proprioceptive Sense	60.7 (7.4)	54.1 (9.7)	<0.001
Factor 1: Sensory Seeking Behaviour	140.3 (15.7)	125.3 (19.3)	<0.001
Factor 2: Sensory Under-Responsivity	111.5 (5.4)	105.9 (8.3)	<0.001
Factor 3: Sensory Over-Responsivity	137.3 (9.1)	126.1 (14.7)	<0.001
Factor 4: Stability of Sensory Responsivity	26.3 (4.6)	23.6 (5.0)	<0.001

## Data Availability

The data presented in this study are available on request from the corresponding author. The data are not publicly available due to ethical reasons.

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
