# Peer review of "Psychometric Properties of the Italian Version of Sensory Processing and Self-Regulation Checklist (SPSRC)"

_healthcare, 2022, doi:10.3390/healthcare11010092_

Round 1

Reviewer 1 Report

I believe this translation and factor analysis of a sensory questionnaire is relevant to many providers and researchers as sensory symptoms are increasingly being recognized as components in other disorders (ASD, ADHD) or disorders to themselves (ie, sensory processing disorders). My main criticism is there is minor editing needed for English language clarity and scientific wording. Otherwise, it appears to be thorough and I believe it's a valuable addition to the literature. My particular comments are below. 

-   Introduction

o   Some editing needed for English language clarity.

o   Some informal/non-scientific language in the introduction, including “as a matter of fact” and “two sides of the same coin.”

o   A mix of citation styles used; in some cases, only a number is used, like this.(5) In others, the citation is mentioned by the author’s name like like: Smith 2020 (5). Please edit in alignment with the journal’s formatting requirements for consistency.

·      Methods: editing needed for clarity

o   “Parents of children between 3 and 8,11 years old were recruited.” Unclear what this means. (8,11 may be a different notation style where a comma is used to indicate a decimal point- I think this is the convention in Europe? Unclear)

o   Cross-sectional study, entirely online I think? Please clarify if so. 

o   Subscales of questionnaire should be consistently capitalized

§  Physiological conditions

§  Behaviors When Facing Changes or Challenges

o   “A higher score indicates a more favorable performance (SPSRC)” what does this mean when talking about sensory processing? Higher score means fewer symptoms?

o   “Short Sensory Profile….. sensory stimuli in children ages 3-14,11 years.” Unclear what this means, see earlier comment about decimal point. 

o   “A higher score indicates a more favorable performance (SSP-2”). What does this mean in the context of sensory symptoms? Fewer symptoms?

o   Table 1

§  Please include number in Table 1 with both number and percent (n (%)), for example, for number of male participants included in the study.

§  Organization is confusing, ie, having Residence (North, Centre, South) all in one row. Could have Residence as a header and then rows with each of the residence areas separately ie

§  Residence

·      North

·      Centre

·      South

·      Results

Mean age was reported as 5,7 (assuming this is 5.7 in US notation) with “DS” in parenthesis, was this supposed to be SD for standard deviation? Please define.

§  Table 2, please include numbers as n (%)

o   Structural validity

§  Confirmatory factor analysis

§  Bartlett’s test of sphericity was significant; please clarify what this means (I believe it means that matrix elements are correlated, which is required when performing a CFA)

Reviewer 2 Report

The article presents the results of the psychometric evaluation of the Italian Version of Sensory Processing and Self-Regulation Checklist (SPSRC). The samples of 30 children with developmental disabilities (DD) and 312 with Italian typically developing (TD) children are compared. The structural validity test indicates that the entire sample was used. It is necessary to clarify - "all" including 30 children with developmental disorders or without their data.

Direct and reverse translations were reasonably matched against the original Chinese SPSRC and English versions. As a result, a questionnaire was obtained for evaluators of children in Italian. After CFA, out of 130, the authors reasonably (criterion 0.35) left 127 points.

In addition to the Italian version of the Checklist (SPSRC), The Short Sensory Profile 2 with 34-items parent questionnaire was also used.

In addition to structural validity, discriminant validity was also assessed, taking into account the age and gender of children.

Psychometric indicators show good reliability and validity of the Italian version of the Checklist (SPSRC). The data are generally correct. Although, it seems to me, the use of the Pearson coefficient is not quite suitable for data on Likert scales, and it would be more adequate to apply a non-parametric criterion (Spearman or others).

At the same time, the substantive aspects of the tested methodology should be clarified. Thus, it is required to give examples of the lack of orientation of children to sensory stimuli. The authors directly introduce a reference to the processes of attention as being regulated by external environmental stimuli. At the same time, it is rightly said about self-regulation; the question remains how the relationship between these concepts is understood meaningfully. Further. The Discussion states that the regulatory functions of attention in response to stimuli follow in their development the development of emotion regulation. I would like to clarify with what methods of statistical evaluation the authors associate this. A counter-hypothesis is meaningfully possible: external attention leads to the development of control over emotions and, accordingly, an increase in the level of self-regulation.

Without expanding the context of the variables taken into account and changing the position of the child with age, the resulting gender differences are interpreted unconvincingly as different trajectories of spontaneous development for boys and girls. This is not so, since the requirements for emotional regulation for boys are somewhat different, this is regulated by parents and society. For example, it is less permissible for a boy to cry (he hears that he is a man in the future, men do not cry).

Thus, while generally having no objections to the psychometric assessment of the Italian version of the SPSRC, it is necessary to provide deeper justifications and hypotheses, and an understanding of the relationship between emotional self-regulation and sensory attention.otional self-regulation and sensory attention.
